# Ultra-Short Polarization Rotator Based on Flat-Shaped Photonic Crystal Fiber Filled with Liquid Crystal

**DOI:** 10.3390/ma15217526

**Published:** 2022-10-27

**Authors:** Rui Liu, Tiesheng Wu, Yiping Wang, Zhihui Liu, Weiping Cao, Dan Yang, Zuning Yang, Yan Liu, Xu Zhong

**Affiliations:** 1Guangxi Key Laboratory of Wireless Broadband Communication and Signal Processing, School of Information and Communication, Guilin University of Electronic Technology, Guilin 541004, China; 2Key Laboratory of Optoelectronic Devices and Systems of Ministry of Education and Guangdong Province, College of Optoelectronic Engineering, Shenzhen University, Shenzhen 518060, China; 3Guangdong and Hong Kong Joint Research Centre for Optical Fibre Sensors, College of Optoelectronic Engineering, Shenzhen University, Shenzhen 518060, China

**Keywords:** polarization rotator, photonic crystal fiber, liquid crystal, FEM

## Abstract

In this study we demonstrate a high-performance polarization rotator (PR) based on flat-shaped photonic crystal fiber. The flat surfaces of the fiber are plated on gold films as electrodes, and the core of the structure is filled with liquid crystal. The polarization rotation characteristics of the flat-shaped fiber can be effectively adjusted by applying external voltage. The optical properties are analyzed using the finite element method (FEM). The results show that the magnitude of the modulation voltage is closely related to the thickness of the flat fiber. When the fiber thickness is 20 μm, only 100 V is required to achieve the highest PR performance. In the wavelength of the 1.55 μm band (~200 nm bandwidth), the conversion length of the PR is only 3.99 μm, the conversion efficiency is close to 100%, and the minimum crosstalk value is −26.2 dB. The presented PR, with its excellent performance, might enable promising applications in the communication system and the photonic integrated circuits.

## 1. Introduction

With the development of photonic integrated circuits (PICs) in recent years, polarization rotators (PRs) that can eliminate the polarization dependence of PICs have also become a research hotspot [1,2,3]. At the same time, the compactness and stability of the PICs determine that the PR also needs to have these characteristics [4]. Therefore, research on high-performance, highly compact PRs is necessary. Normally, there are two schemes for designing a PR, i.e., active and passive PRs devices. The passive scheme controls the mode evolution by changing the geometry of the waveguide [5,6,7,8], but this method has the issues of unidirectional conversion and large volume. The active scheme uses mode coupling [9,10,11,12,13,14]. The main design idea is to change the birefringence of the waveguide so that the input light can excite two mutually orthogonal polarization modes on the transmission section. PRs can be achieved due to the different propagation constants of the two polarization modes.

Photonic crystal fiber (PCF) is widely favored by scholars due to the design freedom of the arrangement of the air holes and their stability. It is a simple solution to break the symmetry of the PCF by changing the distribution of the air holes in the PCF in order to achieve high birefringence [13,15,16]. However, it is not enough to rely on the birefringence produced by changing the distribution or shape of the stomata, because the conversion length of this method is generally too long. Thus, researchers thought of filling high birefringence materials into photonic bandgap PCFs. In 2011, M. F. O. Hameed et al. designed and verified that a polarization rotator can be realized by filling liquid crystal into a soft glass PCF [11]. In 2013, their team designed an ultra-short PR that filled the liquid crystal into the commonly used hexagonal PCF [17]. This PR achieves an ultra-compact device length of 4.085 μm and a polarization conversion rate of almost 100%. In 2017, Lin Yu et al. [18] designed an ultra-short polarization rotator assisted by liquid crystal based on spiral PCF in order to achieve a conversion length of 4.17 μm and crosstalk of −20.93 dB. Although filling the liquid crystal into the PCF greatly reduces the conversion length and improves the conversion efficiency, regrettably, the size of the fiber in these research and design schemes is very small, which causes difficulties in actual manufacturing. Moreover, in different usage scenarios, we expect our design to be able to change the performances through external excitation without changing the structure. In 2011, Alexandros K. Pitilakis et al. proposed the use of soft glass as a substrate and air-hole filled liquid crystals for polarization converter design [19]. The structure they designed deposits electrodes that control the angle of the liquid crystal molecules in the cladding of the fiber and requires a voltage of 150 V to achieve the best performance of the fiber. Obviously, the voltage of this method is still very high. Therefore, in order to make the modulation conditions easier to achieve, another structure needs to be designed.

In this work, we propose an ultra-short polarization rotator based on a flat-shaped PCF. The flat surfaces of the proposed structure are coated with thin metal films to constitute electrodes. The core is filled with liquid crystal, and a tight ring of air holes is added around the liquid crystal core to increase birefringence by adjusting the voltage to control the birefringence of the liquid crystal and thereby achieve the modulation of the rotator. In order to obtain more accurate results, the finite element simulation software COMSOL was used to analyze the designed structure. The suggested structure at a modulation voltage of 100 V can achieve a conversion efficiency close to 100%, and the conversion length is reduced to 3.99 μm. Meanwhile, the crosstalk is also reduced to −26.2 dB. Compared with the existing research, the proposed PR yields high performance improvement and can realize the voltage control of the performance; the modulation voltage is effectively reduced also, and the fabrication of optical fibers is relatively easy.

## 2. Materials and Methods

The cross-section of the proposed flat-shaped PCF is shown in Figure 1. Gold is plated on the top and bottom surfaces of the fiber as electrodes to achieve voltage regulation of the birefringence of the liquid crystal core.

The substrate material of the PCF is silica and its refractive index is given by the Sellmeier formula [20]:(1)$$n{\left(\lambda \right)}^{2}=1+\frac{A_{1}\lambda^{2}}{\lambda^{2}-B_{1}}+\frac{A_{2}\lambda^{2}}{\lambda^{2}-B_{2}}+\frac{A_{3}\lambda^{2}}{\lambda^{2}-B_{3}}$$
where *A*_1_, *A*_2_, *A*_3_, *B*_1_, *B*_2_, and *B*_3_ are 0.6961663, 0.4079426, 0.8974794, 0.0684043, 0.1162414, and 9.896161, respectively.

Furthermore, the thickness of the fiber is h, and the upper and lower planes of the fiber are coated with 40 nm gold thin films as electrodes. The choice of gold can avoid the oxidation of the electrode in the air. In order to increase the birefringence of the fiber, eight air holes with a radius of r_0_ are introduced around the core, and the distance between each small air hole and the core is D_0_. The liquid crystal inserted into the core is the E7 liquid crystal. E7 liquid crystal has two refraction characteristics, namely ordinary refraction no and extraordinary refraction ne. Its refractive index is given by the following Cauchy formula:(2)$$n_{\left(o,e\right)}=A_{\left(o,e\right)}+B_{\left(o,e\right)}\lambda^{-2}+C_{\left(o,e\right)}\lambda^{-4}$$
where *A_o_*, *B_o_*, and *C_o_* are the coefficients of the Cauchy model. Their values are related to temperature, and when the temperature is 25 °C, *A_o_* = 1.4994, *B_o_* = 0.0070 μm^2^, *C_o_* = 0.0004 μm^2^, *A_e_* = 1.6933, *B_e_* = 0.0078 μm^2^, and *C_e_* = 0.0028 μm^2^, respectively. The dielectric tensor of the liquid crystal is [21]:(3)$$\varepsilon_{r}=\left[\begin{array}{ccc} n_{o}^{2}\sin^{2}\phi +n_{e}^{2}\cos^{2}\phi & \left(n_{e}{}^{2}-n_{o}{}^{2}\right)\cos\phi \sin\phi & 0\\ -\left(n_{e}{}^{2}-n_{o}{}^{2}\right)\cos\phi \sin\phi & n_{e}{}^{2}\sin^{2}\phi +n_{o}{}^{2}\cos^{2}\phi & 0\\ 0 & 0 & n_{o}{}^{2} \end{array}\right]$$
where *ϕ* is the rotation angle of the liquid crystal molecules which is modulated by the deflection voltage [22]:(4)$$\varphi =\{\begin{array}{cc} 0{}^{\circ} & V\leq V_{c}\\ 90{}^{\circ}-2\tan^{-1}\left[\exp(-\frac{V-V_{c}}{30V_{c}})\right] & V>V_{c} \end{array}$$
where *V_c_* is the threshold voltage and its value is: *V_c_* = (π/2R) √(k_11_/Δε); *V_c_* is only related to material properties, where the elastic constant k_11_ = 34 pF, anisotropic dielectric constant Δε = 10, and R = 1/2 Λ. Λ is the fiber stomatal pitch and its value in this design is 7.8 μm. It can be calculated that *V_c_* is 0.743 V.

The polarization rotator designed in this paper is realized by mode coupling. When the incident light is TE mode or TM mode, due to the different propagation constants of the two modes, a phase difference will occur between the excitation modes. When the phase difference reaches π, a single mode can be obtained at the output end to achieve polarization rotation. Figure 2 is a schematic diagram of the quasi-TE mode and TM mode.

The length that produces the π phase difference is defined as the conversion length, and its magnitude is given by the following formula:(5)$$L_{\pi }=\frac{\pi }{\beta_{TE}-\beta_{TM}}=\frac{\lambda }{2\left(n_{TE}-n_{TM}\right)}$$
where *β_TE_* and *β_TM_* are the propagation constants of the quasi-TE mode and the quasi-TM mode, respectively; *n_TE_* and *n_TM_* are the real part of the effective refractive index.

The modal hybridness is an important parameter of the polarization rotator. It represents the energy distribution of the two modes. When the modal hybridness is 1, it means that the energy distribution of the two modes is uniform. At this time, the performance of the rotator is the best. Defined as the double integral of the electric field components of the two modes [17]:(6)$$S=\iint \left(E_{x}(\mathrm{TE},1\right)E_{x}(\mathrm{TM},2)+E_{y}(\mathrm{TE},1)E_{y}(\mathrm{TM},2))dxdy$$

Flat-shaped PCFs can be manufactured by processing the commonly used triangular lattice fibers. With the improvement of fiber fabrication process, the fabrication of complex structure fiber is possible. In 2008, Y.Y. Wang et al. [23] reported a method of preparing annular pores around the fiber core by chemical etching. In addition, the selective permeation and filling of liquid crystal into the PCF has been realized [24]. Nowadays the preparation process of PCF is relatively mature [25]. These reports and the current fabrication process of commercial PCF provide theoretical and practical support for the fabrication of our proposed fibers.

## 3. Results

In this design, the thickness of the fiber is 20 μm, the diameter of the liquid crystal core r, the cladding air holes r_1_, and the pores around the liquid crystal core r_0_ are taken as 8 μm, 3.9 μm, and 4 μm, respectively. The cladding air holes interval is Λ = 7.8 μm, and the distance between each pore and the core is D_0_ = 0.2 μm; when the operating wavelength is 1.55 μm, the background refractive index is 1.444.

Since the fiber base material is silica, which is a non-polarized material, the thickness of the fiber can affect the potential difference between the two ends of the liquid crystal, thereby affecting the rotation angle of the liquid crystal molecules (*ϕ*). Therefore, the impact of the thickness of the fiber on the potential difference between the two ends of the liquid crystal core is first studied, as shown in Figure 3. It can be seen that when the same voltage is applied to the electrode, the thinner the thickness, the greater the potential difference between the two ends of the liquid crystal. This conforms to the theoretical conjecture. Considering the possibility of actual manufacturing, the thickness is selected as 20 μm.

Modal hybridness and conversion length are related to the rotation angle of liquid crystal molecules (*ϕ*); we control *ϕ* by applying voltage. Therefore, the relationship between the modal hybridness and the conversion length and the potential difference between the two ends of the liquid crystal at a wavelength of 1.55 μm and 25 °C was studied. As shown in Figure 4, when the potential difference is between 0–20 V, the modal hybridness gradually increases to the maximum as the potential difference increase correspondingly, the *ϕ* is increased from 0° to 45° at this time. However, as the potential difference gradually increases, the modal hybridness gradually decreases from the maximum until the potential difference reaches the saturation voltage of the liquid crystal, and the rate of decline gradually slows down; at this time, the *ϕ* gradually increased from 45° to 90°, and the rate of increase gradually decreased. Studies have shown that the modulation of the polarization rotator can be achieved by applying a voltage to the electrode.

Compared with the conversion length and modal hybridness, the conversion ratio (Px) can better reflect the performance of the polarization rotator. It is defined as the ratio of the power of the output TM mode to the power of the input quasi-TE, and it is related to the *ϕ* and the conversion length, as shown in Figure 5.

As the conversion length gradually increases from 0 μm to Lπ, the Px gradually increases from 0 to the maximum. When the *ϕ* is 45°, the maximum Px is close to 1. When the *ϕ* is 0°, 10°, 20°, 30°, and 40°, the Px is the same as 90°, 80°, 70°, 60°, and 50°. This property corresponds to the degree of modal hybridness. At the same time, the conversion length in Figure 4 is also related to the *ϕ*, and its law is consistent with Figure 3. Moreover, it can be seen from Figure 4 that if the fiber is long enough, the Px changes periodically; when the *ϕ* is 45° and the fiber length is 3.99 μm, the Px is still close to 1.

Another important parameter of PR is crosstalk. Figure 6 is a graph of crosstalk under different potential differences. It can be seen that between 0–20 V, the crosstalk value gradually decreases, at this time, the *ϕ* is between 0–45°. While the voltage is 20 V, the crosstalk value is the lowest, reaching −26.2 dB. This is because at 20 V, the mode mixing degree is maximum, and changing the voltage at this time will obviously change the crosstalk value.

The molecular density of the liquid crystal is different at different temperatures so that it will affect the refractive index of the liquid crystal. Figure 7a is a graph of the conversion length and modal hybridness at different temperatures. It can be seen that as the temperature rises, the modal hybridness and conversion length both rise. This is due to the fact that the extraordinary refractive index ne of the liquid crystal decreases from 1.7096 to 1.6438 as the temperature rises from 15–50 °C in the 1.55 μm band; its ordinary refractive index decreases from 1.5034 to 1.5017 from 15–35 °C, and its no increases from 1.5017 to 1.5089 as the temperature rises from 35–50 °C. From the law of change of liquid crystal birefringence, we can see that the birefringence effect becomes smaller as the temperature rises, and the conversion length of PR is inversely related to birefringence, so it makes the conversion length gradually become larger as the temperature rises [26]. Although the mixing degree of the rotator will increase with the increase of temperature, it can be seen from Figure 7b that when the fiber length is 3.99 μm instead of Lπ, the crosstalk value and conversion ratio are the same as Lπ at 25 °C. As the temperature rises, the crosstalk value gradually rises, and the conversion ratio gradually decreases. However, in the manufacturing process, due to the existence of errors, it is impossible to make the fiber length Lπ very accurately. It can be seen that the fiber designed in this paper can tolerate an error of ±5 °C at 3.99 μm.

Next, the influence of fiber parameters on conversion ratio, conversion length, and crosstalk is studied, as show in Figure 8. First, the influence of the core size on the conversion length and modal hybridness is studied. At this time, the temperature is maintained at 25 °C, the voltage across the liquid crystal is 20 V, the distance between the pores around the core and the core is 0.2 μm, and the pore radius is 2 μm. Figure 8 shows the relationship between the conversion length and the modal hybridness and the size of the core. As the core size increases, the modal hybridness remains basically unchanged. When the core radius exceeds 4 μm, it decreases slightly. This is because, when the core size is too large, a small part of the fundamental mode will be converted to higher-order modes. The conversion length gradually decreases from 4.04 μm to 3.98 μm, because, as the core size increases, the birefringence between the two modes will increase.

The duty cycle of the air holes around the core also affects the birefringence of the core. Considering that the size of the pores around the core and the distance between the pores and the core will affect the duty cycle, we keep the temperature at 25 °C and the core radius at 4 μm, the effects of pore spacing and pore size on conversion length, conversion ratio, and crosstalk were studied, respectively. Figure 9a shows the effect of the size of the air hole on the conversion length and conversion ratio when keeping the distance between the air pores and the core (D_0_) at 0.2 μm. When the air hole radius increases from 1 μm to 1.3 μm, the conversion length remains unchanged at 3.9948 μm. When the air holes radius continues to increase, the conversion length will rapidly drop to 3.9928 μm. Although the conversion length has changed, the change is not obvious, but as the air hole radius increases, the conversion efficiency gradually increases from 0.993 to close to 1. Figure 9b shows that the fixed air hole size is 2 μm; when the air hole and the fiber core are at different intervals, the graph of crosstalk and conversion rate when the fiber length is Lπ and 3.99 μm. As the interval gradually increases from 0.2 μm to 1.2 μm, the conversion ratio gradually decreases, and the crosstalk gradually increases. Moreover, with the increase of the interval, the crosstalk value and conversion ratio of the optical fiber taking 3.99 μm and Lπ are only close to the same when 0.2 μm, and then the difference will become larger and larger. This can be explained as the equivalent refractive index near the liquid crystal core is relatively low when the surrounding air holes have a relatively large occupancy. At this time, the birefringence effect of the liquid crystal is more obvious.

Considering that the refractive index of the material is related to the wavelength, the influence of the wavelength on the Px and crosstalk value is studied, as shown in Figure 10. When the wavelength is between 1.2 μm and 1.55 μm, the Px gradually rises to the maximum value, and the crosstalk value gradually decreases to the minimum value. When it exceeds 1.55 μm, the Px will decrease, and the crosstalk value will increase. Between 1.25 μm and 1.45 μm, the Px of 3.99 μm for fiber is significantly lower than the value when the fiber is Lπ; when the wavelength exceeds 1.45 μm, the two are almost the same. At the same time, when the fiber length is 3.99 μm, the crosstalk value after the optical wavelength is 1.45 μm is lower than −20 dB. The operating wavelength of the optical fiber is 1.55 ± 0.1 μm.

## 4. Conclusions

In conclusion, we propose and demonstrate a new type of PR with an ultra-short conversion length. The proposed structure is made of flat-shaped PCF filled with liquid crystal in the core. We investigated the influence of structural parameters on the polarization conversion performance. The fiber thickness can effectively reduce the modulation of flat-shaped PCF voltage of PR; the maximum conversion ratio can be achieved at a voltage of just 100 V. Its conversion length is just 3.99 μm, and the working crosstalk is as low as −26.2 dB. Under the optimal parameters, the PR can offer an almost 100% polarization-conversion ratio with a wavelength of 1.55 ± 0.1 μm and a temperature tolerance of 5 °C. The basic size of the fiber is close to the commercial size, and this opens up the possibility of manufacturing. The proposed PR has great application potential in actual manufacturing and modern communication systems.

## Figures and Tables

**Figure 1 materials-15-07526-f001:**
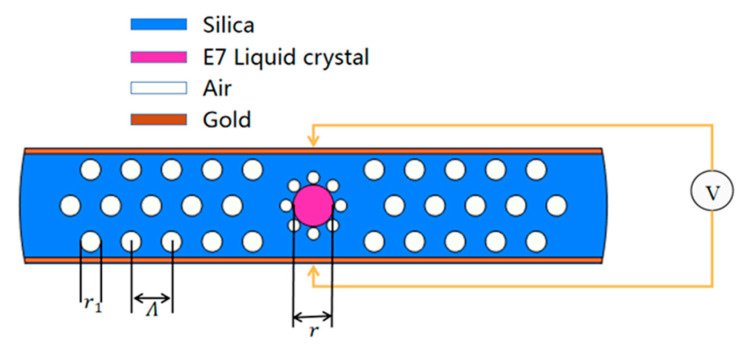
Suggested flat-type PCF.

**Figure 2 materials-15-07526-f002:**
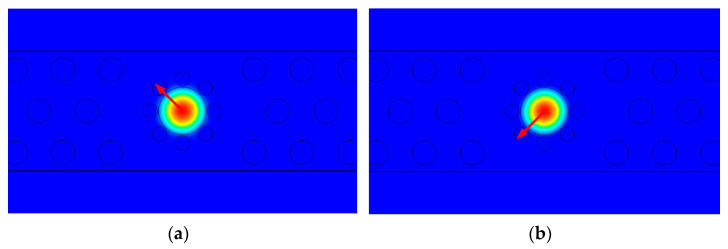
(**a**) Electromagnetic field distribution of the quasi-TE model; (**b**) electromagnetic field distribution of the quasi-TM model.

**Figure 3 materials-15-07526-f003:**
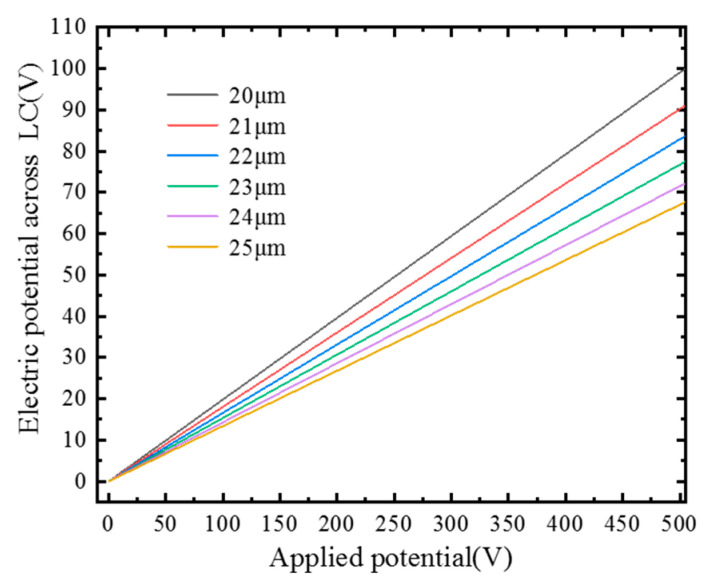
The relationship between the potential difference between the two ends of the liquid crystal with different thicknesses of the fiber and the applied voltage.

**Figure 4 materials-15-07526-f004:**
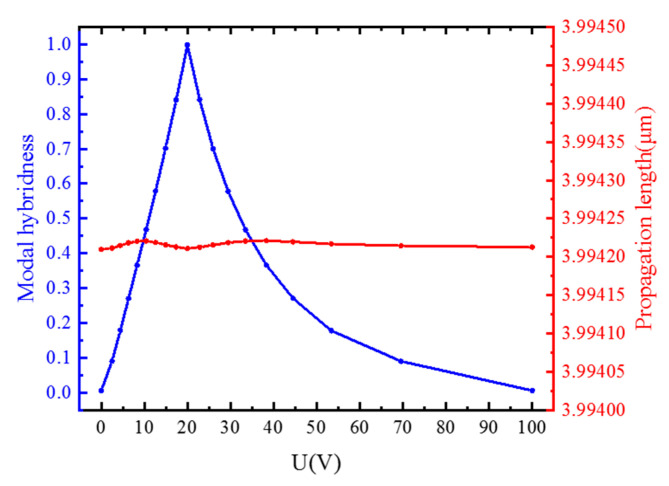
The relationship between the modal hybridness and the conversion length with the potential difference between the two ends of the liquid crystal.

**Figure 5 materials-15-07526-f005:**
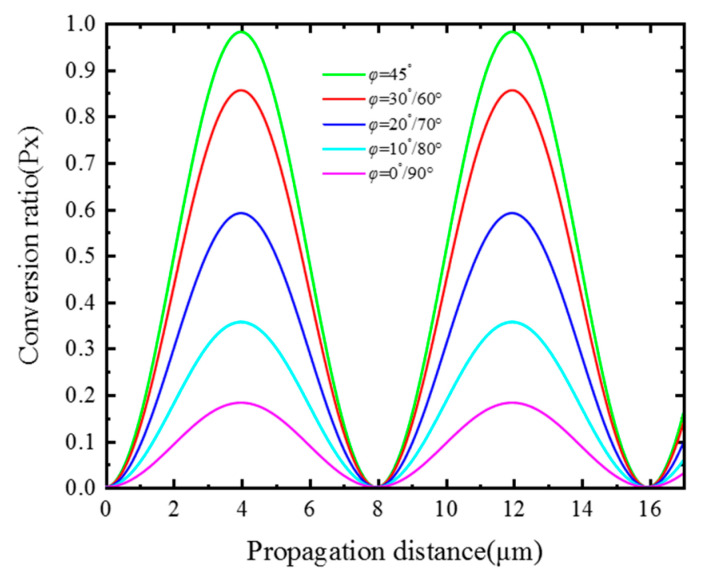
The conversion ratio changes with the propagation distance under different liquid crystal rotation angles.

**Figure 6 materials-15-07526-f006:**
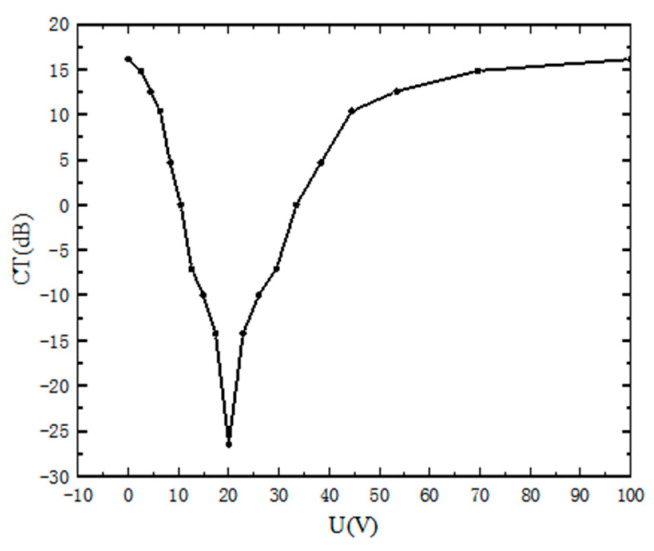
Variations of CT with the potential difference.

**Figure 7 materials-15-07526-f007:**
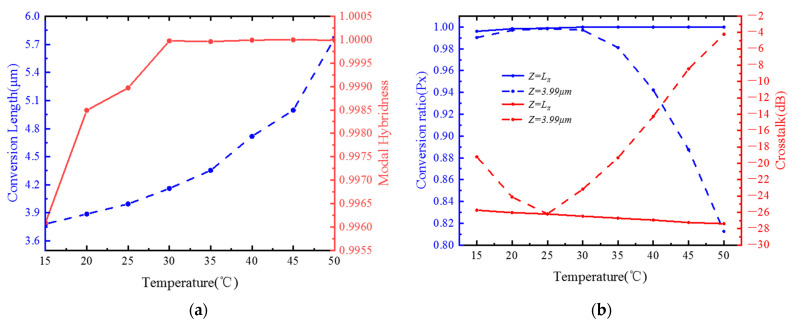
(**a**) Variations of conversion length and modal hybridness with temperature; (**b**) conversion ratio (Px) and CT as functions of temperature at Z = Lπ and Z = 3.99 μm.

**Figure 8 materials-15-07526-f008:**
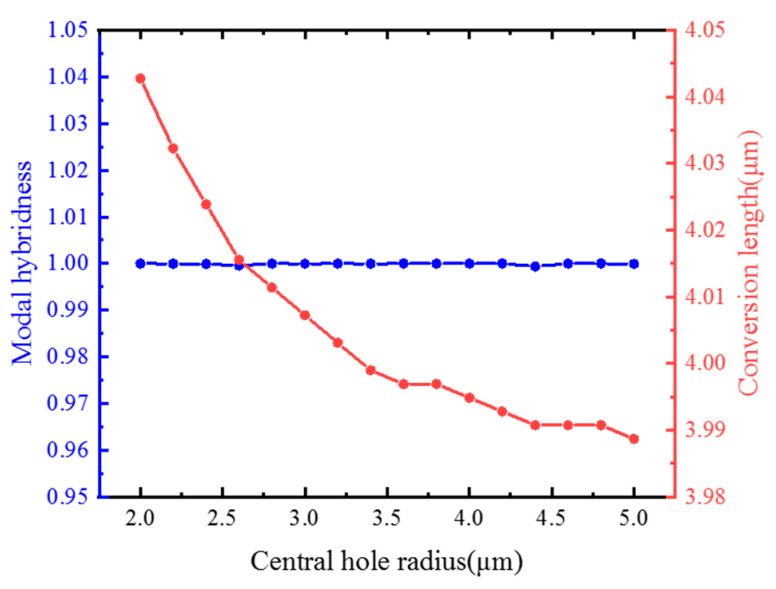
Dependences of the modal hybridness and conversion length for different values of the central hole radius r.

**Figure 9 materials-15-07526-f009:**
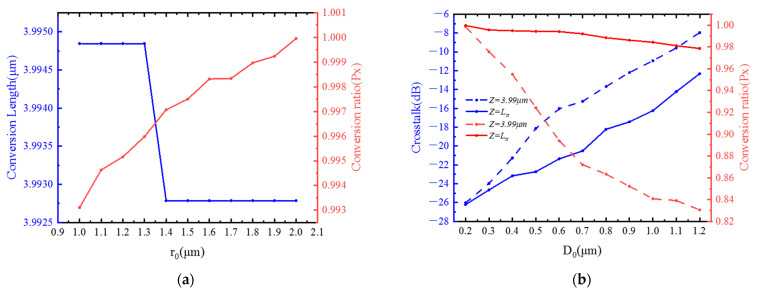
(**a**) Variations of conversion length and conversion ratio with the size of the air hole (r_0_); (**b**) CT and conversion ratio at Z = Lπ and Z = 3.99 μm with different values of the distance between the air pores and the core (D_0_).

**Figure 10 materials-15-07526-f010:**
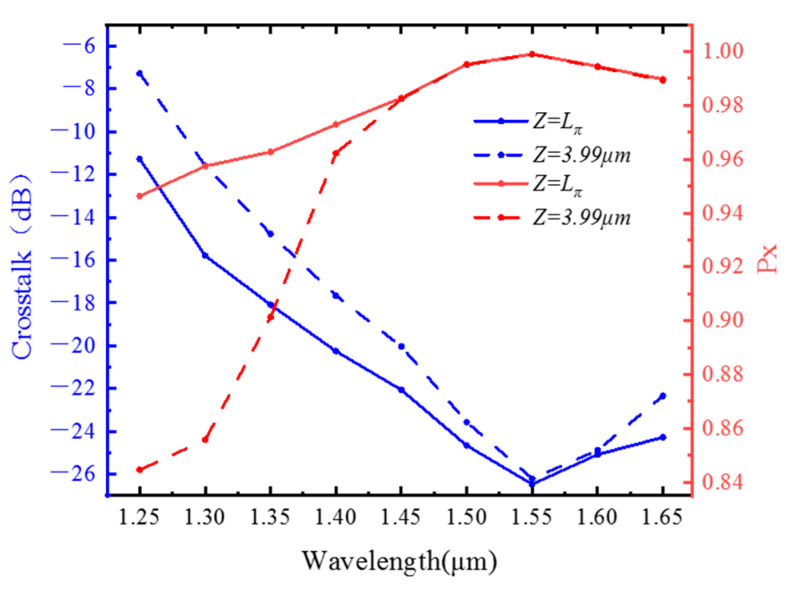
Variations of the crosstalk and conversion ratio at Z = Lπ and Z = 3.99 μm with wavelength.

## Data Availability

Data sharing is not applicable to this article.

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
