# Peer review of "Ultra-Short Polarization Rotator Based on Flat-Shaped Photonic Crystal Fiber Filled with Liquid Crystal"

_materials, 2022, doi:10.3390/ma15217526_

Round 1

Reviewer 1 Report

1-      Clearly state the advantages of the proposed flat-shaped PCF. Please provide you strategy design such a structure. Because so many other structure may be proposed.

2-      Are you used a software of wrote a code. Please specify. If you wrote a code, adding a small section about it may be helpful. Otherwise please write the software name.

3-      You selected e.g. "thickness of the fiber is 20 μm". How are you selected this and other parameters value. Arbitrarily? Or they are optimized?

Author Response

  1. Response to comment: Clearly state the advantages of the proposed flat-shaped PCF. Please provide you strategy design such a structure. Because so many other structure may be proposed.
    Response: It is really true as Reviewer suggested that we did not describe the idea of the designed structure very well. For this problem, we added in the introduction section the advantages of designing a flat-shaped PCF and why we proposed such a structure. And the additional part is marked in red in the original 56 to 62 sentences.
  2. Response to comment: Are you used a software of wrote a code. Please specify. If you wrote a code, adding a small section about it may be helpful. Otherwise please write the software name.
    Response:
    We are very sorry for our negligence of we don't specify the simulation software used. In this study, we use the finite element method analysis software COMSOL to simulate and optimize the proposed structure. Considering the Reviewer’s suggestion, we have added clarification in sentences 67 to 69 in the article.
  3. Response to comment: You selected e.g. "thickness of the fiber is 20 μm". How are you selected this and other parameters value. Arbitrarily? Or they are optimized.Response: Many thanks to the Reviewers for the questions we asked. For this problem, in the first part of the article Results, we investigated the relationship between the thickness of the fiber and the voltage, and the results showed that the thinner the fiber, the lower the modulation voltage needed. At the same time, we cannot reduce the thickness of the fiber indefinitely, considering that the structure of the fiber core is not destroyed. We choose 20um as the final optimization result.

Special thanks to you for your good comments.

Reviewer 2 Report

In this manuscript, Liu et al. reports a new type of polarization rotator by filling liquid crystal in the core of flat-shaped photonic crystal fiber. Such polarization rotator exhibits ultra-short conversion length and nearly 100% conversion efficiency. This paper is complete and has clear logic. It can be considered for the publication in this journal after revising the following minor issues:

1. Page 2, line 74-75: The Sellmeier formula shows the relationship between refractive index and wavelength, but the authors describe “…and its dielectric constant is given by the Sellmeier formula [21]:”.

2. Page 3, Equation (4): Please unify the symbol of the rotation angle of LC molecules.

3. Page 6, line 179-182: “This is because as the temperature rises, the extraordinary refractive index ne of the liquid crystal molecules changes more obvious, during the ordinary refractive index no has a relatively small change, and its birefringence effect is proportional to the temperature.” This is not accurate and clear. Please refer to some papers for description.

4. Page 7, line 212-213: As the authors describe “When the air hole radius increases from 1μm to 1.4 μm, the conversion length remains unchanged at 3.9948 μm”, but from Figure 9(a), when the air holes radius is 1.4 μm, the conversion length already decreases to 3.9928 μm.

Author Response

  1. Response to comment: Page 2, line 74-75: The Sellmeier formula shows the relationship between refractive index and wavelength, but the authors describe “…and its dielectric constant is given by the Sellmeier formula [21]:”

Response: We are very sorry for our negligence of this little issue. We have made correction according to the Reviewer’s comments.

  1. Response to comment: Page 3, Equation (4): Please unify the symbol of the rotation angle of LC molecules.

Response: We are very sorry for our negligence. We have added to this part according to the Reviewer’s suggestion.

3.Response to comment: Page 6, line 179-182: “This is because as the temperature rises, the extraordinary refractive index ne of the liquid crystal molecules changes more obvious, during the ordinary refractive index no has a relatively small change, and its birefringence effect is proportional to the temperature.” This is not accurate and clear. Please refer to some papers for description.

Response: We are very sorry for our negligence. We have re-written this part according to the Reviewer’s suggestion, and the relevant reference was added.

4.Response to comment: Page 7, line 212-213: As the authors describe “When the air hole radius increases from 1μm to 1.4 μm, the conversion length remains unchanged at 3.9948 μm”, but from Figure 9(a), when the air holes radius is 1.4 μm, the conversion length already decreases to 3.9928 μm.

Response: We are very sorry for our negligence. We have revised this section based on the Reviewer’s suggestions.

Round 2

Reviewer 1 Report

there is no comment.